

# A small target detection algorithm based on improved YOLOv5 in aerial image

PengLei Zhang* and Yanhong Liu*

Software College, Henan University, KaiFeng, Henan, China
* These authors contributed equally to this work.

## ABSTRACT

Uncrewed aerial vehicle (UAV) aerial photography technology is widely used in both industrial and military sectors, but remote sensing for small target detection still faces several challenges. Firstly, the small size of targets increases the difficulty of detection and recognition. Secondly, complex aerial environmental conditions, such as lighting changes and background noise, significantly affect the quality of detection. Rapid and accurate identification of target categories is also a key issue, requiring improvements in detection speed and accuracy. This study proposes an improved remote sensing target detection algorithm based on the YOLOv5 architecture. In the YOLOv5s model, the Distribution Focal Loss function is introduced to accelerate the convergence speed of the network and enhance the network's focus on annotated data. Simultaneously, adjustments are made to the Cross Stage Partial (CSP) network structure, modifying the convolution kernel size, adding a new stack-separated convolution module, and designing a new attention mechanism to achieve effective feature fusion between different hierarchical structure feature maps. Experimental results demonstrate a significant performance improvement of the proposed algorithm on the RSOD dataset, with a 3.5% increase in detection accuracy compared to the original algorithm. These findings indicate that our algorithm effectively enhances the precision of remote sensing target detection and holds potential application prospects.

# INTRODUCTION

In recent years, uncrewed flight technology has matured significantly, leading to notable advancements in uncrewed aerial photography. This technology is now used extensively in both military and civilian applications (*Jiao et al., 2019*). Compared to traditional aerial photography or manual survey methods, UAV aerial photography offers several advantages. It is a cost-effective solution, as it does not require expensive equipment or extensive human resources. UAV aerial photography also provides high-quality remote sensing data, which is critical for decision-makers. Furthermore, it offers real-time monitoring capabilities and instant feedback, enabling quick access to panoramic images and video data of ground conditions, which facilitates rapid decision-making and strategic adjustments. However, target detection algorithms, which are closely linked to UAV aerial photography technology, face certain challenges. In complex aerial photography scenarios, there may be a multitude of small targets to identify, primarily due to shooting distance.

Corresponding author
Yanhong Liu,
liuyanhongxmu@163.com

This presents challenges in accurately identifying small targets and achieving real-time detection speed. Therefore, current research is mainly focused on improving target detection algorithms to adapt to these complex situations.

In summary, the development of UAV aerial photography technology holds great promise across various domains. However, continuous refinement of target detection algorithms is imperative to overcome the challenges involved. This is essential for enhancing the accuracy of recognizing small targets and achieving real-time detection speeds, better aligning with the diverse needs of various applications.

Traditional target detection algorithms, such as the R-CNN series (*Girshick et al., 2014*), the SSD series (*Redmon et al., 2016*), and the YOLO series (*Liu et al., 2016*), have proven their efficacy in general detection scenarios. However, they face inherent limitations in achieving highly accurate detection of small targets in complex settings. To overcome these challenges, *Lin et al. (2017a)* introduced a groundbreaking structure within the Feature Pyramid Network (FPN) algorithm. This innovative structure blends bottom-up and top-down feature transfer methods, enabling the fusion of deep feature maps with high semantic content from the upper layers and high-resolution shallow feature maps from the lower layers. This fusion empowers independent predictions, significantly enhancing the effectiveness of target detection algorithms, particularly for small targets. *Kisantal et al. (2019)* harnessed oversampling amplification to boost the detection accuracy of neural networks for small targets. Meanwhile, *Fu et al. (2017)* incorporated inverse convolution in SSD to integrate contextual information, thereby enhancing the readiness of detecting small targets. In *Wang et al. (2019)*, an innovative enhancement was introduced to upgrade the SSD network based on FPN, leading to notable improvements in both speed and accuracy of target detection. In 2019, the advent of the CornerNet (*Law & Deng, 2018*) algorithm ushered in a new era of anchorless frame algorithms. This approach successfully surmounted challenges associated with manual design, inefficient training and prediction processes, and the imbalance of positive and negative samples inherent in anchor frame-based methods. Additionally, CornerNet achieved accuracy on par with anchor frame-based algorithms. *Zhu et al. (2021)* introduced the CBAM attention mechanism into the YOLOv5 model to combat target blurring issues in aerial images. *Miao, Yu & An (2022)* leveraged a multi-level fusion structure to generate multi-scale feature maps with precise location information and semantic features. The generation of these feature maps further refined the scale of candidate regions, enhancing the accuracy of detecting multi-scale aircraft targets in remote sensing images. *Wang et al. (2023)* and *Zhu et al. (2021)* have successfully incorporated the Transformer and feature pyramid structure into the realm of remote sensing image target detection. This integration not only bolsters the network's feature representation capabilities but also intensifies the fusion of multi-scale features. These improvements not only mitigate interference from complex backgrounds but also facilitate deeper extraction of target feature information, ultimately leading to a substantial improvement in detection accuracy. However, it is worth noting that the enhanced model places significant demands on hardware performance and can pose implementation challenges.

In this article, we propose a novel and improved algorithm based on YOLOv5s, to tackle the issue of poor accuracy in detecting small targets in remote sensing. Our contributions can be summarized as follows:

1) We introduced the DFL (*Li et al., 2020*) loss function, which significantly enhances the model's precision and robustness in identifying boundary targets by employing specialized treatment of pixels at the image's edge during neural network training. The design of the DFL loss function improves the model's resilience and accuracy in the detection of small targets.

2) We design a novel SSN module to enhance the neural network. This module enhances the model's capacity for representation by incorporating additional convolutional layers, smaller convolutional kernels, and structures such as residual connectivity, thus improving the model's performance by capturing richer and more abstract data features. Notably, substantial progress has been made in the realm of small target detection.

3) We introduce an improved structure for the attention mechanism, effectively facilitating information interaction and fusion across different contexts and optimizing the inherent structure of information at various levels by judiciously assigning weights. This innovation contributes to enhancing the model's perception and inference capabilities.

The remainder of this article is organized as follows. Our small target detection algorithm is introduced and analyzed in detail in the 'Algorithm Design and Analysis' section. The 'Experiment' section evaluates the performance of our algorithm by a large number of contrast experiments based on public datasets and makes relevant discussions. The 'Conclusion' section summarizes this article.

# ALGORITHM DESIGN AND ANALYSIS

In this article, we propose modifications to the YOLOv5s model in three key aspects: enhancing the neck network model, introducing a new loss function, and incorporating a novel attention mechanism. These modifications are aimed at improving the network's feature extraction capability, consequently enhancing the model's detection accuracy. Detailed explanations of these modifications can be found in sections "Neck Network Design", "Loss Function", and "Attention Mechanism", respectively.

## Neck network design

The YOLOv5 series of models can be categorized into four models based on depth and width: YOLOv5s, YOLOv5l, YOLOv5m, and YOLOv5x. Among them, YOLOv5s is the fastest and lightest model, which is very suitable for deployment and use on mobile devices. This article aims to improve the accuracy of detecting small targets based on the YOLOv5 model.

The depth of a convolutional neural network (CNN) plays a crucial role in feature extraction and characterization. However, as the number of layers in the network increases, it often leads to the extraction of high-level features while simultaneously

resulting in a loss of detailed information. This phenomenon is commonly referred to as the "information bottleneck," where deeper networks tend to focus more on abstracting and generalizing features, potentially overlooking subtle differences in the original input. To address this challenge, researchers frequently employ a strategy of fusing deep and shallow layers in neural networks. This fusion introduces feature information from shallow layers into the model, thereby enhancing the network's expressive power. The key advantage of this fusion approach is its ability to capture both high-level abstract features and detailed information, significantly improving the performance and accuracy of the network.

The design of YOLOv5's input plays a crucial role in target detection, and it employs Mosaic data augmentation and rectangular black-edge filling methods to improve the model's performance and robustness. Mosaic data augmentation is an innovation in YOLOv5's input processing. The method provides more complex and diverse scene information to the model by stitching four different images together to form a hybrid image. This design introduces more perspectives and contexts during the training process, which helps to improve the model's adaptability to different scenes and targets. The adoption of Mosaic data augmentation allows YOLOv5 to perform even better when confronted with real-world, complex and varied target scenes.

To handle images of various sizes and scales, YOLOv5 utilizes the rectangular black edge fill method. This method adjusts the image to a square shape by filling the edges of the image with a black border to meet the model's requirement for a uniform input size. This plays an important role in eliminating inconsistencies in image sizes and improves training stability and convergence speed. The use of rectangular black edge filling provides more consistent and manageable inputs to the model, which helps to improve the effectiveness of training.

YOLOv5s backbone network (Backbone) plays a crucial role in target detection and is responsible for extracting rich features from the input image to provide strong support for subsequent network layers. Prior to the v6.0 version of YOLOv5, the backbone network utilized the Focus structure, and two CSP structures were proposed, which laid the foundation for the performance enhancement of the model. The Focus module introduces an innovative slicing operation at the input of the backbone network. This operation slices the high-resolution input image into multiple low-resolution image blocks. Taking a $4 \times 4 \times 3$ input image as an example, a $2 \times 2 \times 12$ feature map is generated by interval sampling and channel splicing. This operation aims to decompose and integrate the image information efficiently, expanding the number of channels to four times the original to provide richer features for subsequent convolutional operations. In YOLOv5s v6.0, the SPPF module replaces the SPP module in the previous version and is placed at the end of the backbone network. The SPPF module inputs the maximum pooling layer with a convolutional kernel size of $5 \times 5$ through serial inputs before feature fusion. Compared to the SPP module, the SPPF module is less computationally intensive and more efficient, while maintaining the same sensory field. In the backbone structure of YOLOv5s, a combination of the Conv module and the C3 module is utilized to achieve this objective. The Conv module primarily handles downsampling and nonlinear

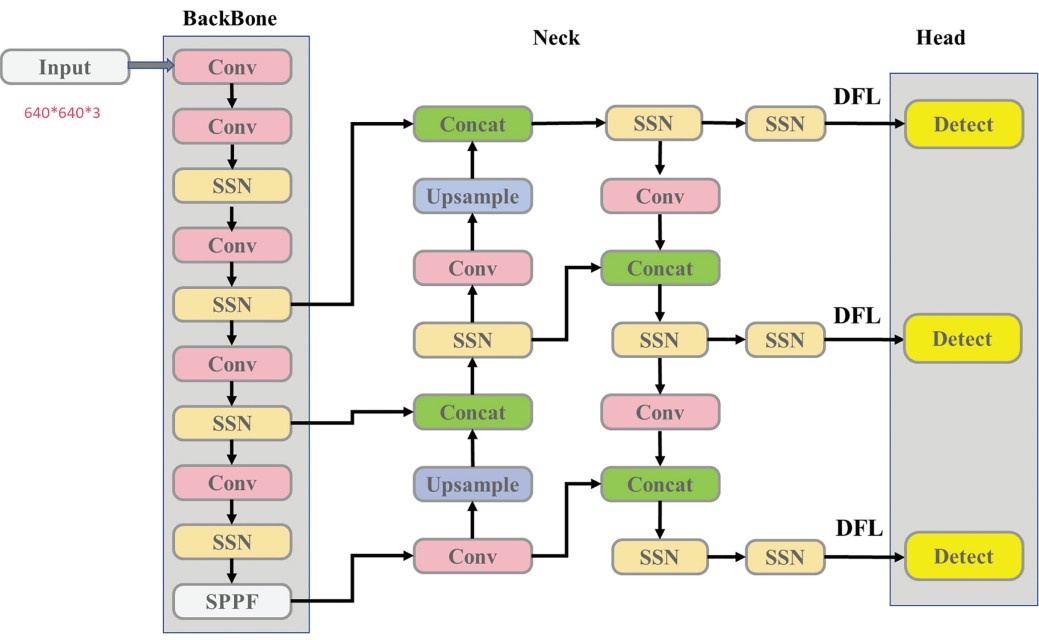

**Figure 1 Our network structure.**

transformation operations to derive more representative and abstracted features, aiding the network in comprehending the overall structure and crucial features within the image. Simultaneously, the C3 module is interposed between Conv modules to bolster feature information for learning residual features. This layout of the C3 module better captures contextual information from the image and leverages the residual structure pioneered by ResNet (*Targ, Almeida & Lyman, 2016*), effectively addressing the issue of gradient vanishing in deep networks and enhancing the stability and reliability of network training and optimization. This design not only heightens the network's sensitivity to detailed information but also improves its capability to express high-level semantic features, thereby enhancing the performance and robustness of YOLOv5s.

In essence, the fusion of deep and shallow layers alongside the utilization of Conv and C3 modules represents a common design strategy in contemporary convolutional neural networks. This strategy effectively overcomes the information bottleneck inherent in deep networks, thereby enhancing feature expression and overall network performance. Finally, the detection head section facilitates fine-grained classification and positional regression of candidate frames, playing a pivotal role in various deep learning tasks not only within computer vision but across diverse domains.

To enhance YOLOv5s's ability to detect small targets in remote sensing applications, we made several modifications to the network structure of YOLOv5s, see Fig. 1. Firstly, we replaced the C3 layer with a new SSN structure (see Fig. 2) and simultaneously optimized the NECK component. The convolution kernel size has been adjusted to 3, improving the extraction of fine details while minimizing the loss of target position information during network transmission. Additionally, during the pre-processing stage, the number of channels has been doubled, and these channels are further divided to incorporate multiple

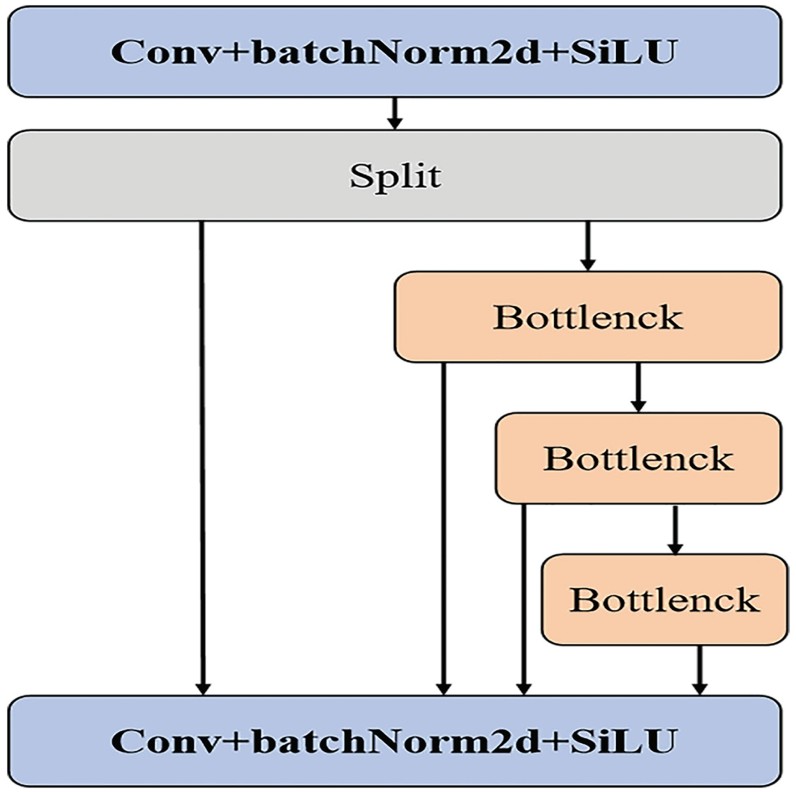

**Figure 2 Stack structure of SSN.**               

stacked modules. Given that most remote-sensing objects are small, this modification addresses the challenges associated with detection. Including multiple stacked modules helps mitigate network degradation and gradient vanishing issues, reducing instances of feature semantic loss and enhancing the network's fusion capabilities. Lastly, a DFL loss function has been introduced to the detection header to improve boundary detection accuracy.

The stacked convolutional module, by concatenating multiple smaller-sized convolutional layers, effectively reduces the parameter and computational load compared to a single large-sized convolutional layer. This design not only maintains model performance but also reduces computational costs, making the model more feasible in resource-constrained environments. Since the stacked convolutional module comprises multiple convolutional layers, each layer can learn features at different scales. Such a design enables the model to better capture multi-scale information of the targets, thereby enhancing detection capabilities across various sizes, particularly for detecting tiny objects.

Through stacked convolutional layers, the model gradually constructs richer and more abstract feature representations. This hierarchical feature extraction and composition aid the model in better understanding image content, thereby improving detection accuracy and robustness. The parameter-sharing characteristic among multiple convolutional layers within the stacked convolutional module means the model can utilize the same weights to

extract features at different positions and scales, further reducing the parameter count and mitigating the risk of overfitting.

As a modular design, the stacked convolutional module can be relatively easily integrated into existing object detection frameworks. By adjusting the number and size of convolutional layers, the complexity and performance of the model can be flexibly controlled to meet the requirements of different scenarios.

## Loss function

YOLOv5s model uses CIoU Loss to calculate the rectangular box loss, there are three main components, the loss of predicting the position of the rectangular box ($L_b box$), the loss of confidence ($L_o bj$), and the loss of categorization ($L_c ls$), the specific formula for CIoU Loss is

$$IoU = \frac{|A \cap B|}{A \cup B} \tag{1}$$

$$GIoU = IoU - \frac{A_c - U}{A_c} \tag{2}$$

$$L = L_{bbox} + L_{obj} + L_{cls}. \tag{3}$$

The IoU loss, commonly used in object detection, measures the intersection over the union between the ground truth box A and the predicted box B, as shown in Eq. (1). Unlike IoU, which focuses solely on the overlapping region, GIoU considers both the overlapping region and other non-overlapping areas to better reflect their degree of overlap, as depicted in Eq. (2). The YOLOv5 loss function consists of three components, as illustrated in Eq. (3).

CIoU Loss incorporates factors such as overlap region, center distance, and aspect ratio of bounding boxes to enhance model training stability and convergence speed. However, it's important to note that while CIoU Loss considers these factors, it may not fully reflect the true differences between the width, height, and confidence of the bounding boxes, which could lead to insufficient precision in regression prediction results.

In YOLOv5s, the confidence loss function employs the BCE Loss, typically used in binary classification problems to measure the difference between the model's predicted class probability and the actual label. For binary classification problems, assuming the model's output is y representing the probability of predicting the positive class, and the actual label is $y_{true}$ (taking values of 0 or 1, representing negative or positive class), BCE Loss is calculated as shown in Eq. (4).

$$BCELoss = -(y_{true} \log(y) + (1 - y_{true}) \log(1 - y)) \tag{4}$$

$y_{true} \log(y)$ is used to measure the loss of the model predicting a positive class when the true label is 1, and $(1-y_{true}) \log(1-y)$ is used to measure the loss of the model predicting a negative class when the true label is 0. Overall, a smaller BCE Loss indicates that the model's prediction is closer to the true label. In the realm of traditional focal loss (*Miao, Yu & An, 2022*), the adjustment of weights for difficult samples is achieved by introducing a focusing parameter (FFP), which prioritizes the handling of challenging-to-classify

samples, thereby improving the model's performance. To address this challenge, this article introduces the Distribution Focal Loss (DFL) loss function, which builds upon the foundations of FL and further takes into account the issue of category imbalance. The fundamental concept behind DFL is to enable the model to provide more precise predictions when dealing with target boundary locations. This means generating sharp predictive probability distributions at clear boundary locations while producing relatively flat predictive probability distributions at ambiguous or uncertain boundary locations. Locations with unclear or uncertain boundaries yield relatively smooth probability distributions. This distinctive feature empowers the model to better adapt to target boundaries in diverse scenarios, ultimately elevating accuracy and robustness in boundary localization.

The specific formula for the DFL loss function is:

$$DFL(S_i, S_{i+1}) = -((y_{i+1} - y) \log(S_i) + (y - y_i) \log(S_{i+1})) \tag{5}$$

$S_i$ and $S_{i+1}$ are the model's predictive confidence for two different categories in target detection. $y_i$ is the corresponding true label, and $y_{i+1}$ is usually the label for the next category. $\log(S_i)$, on the other hand, represents a measure of the difference between the model's predicted probability distributions and the true labels. the goal of DFL is to make the network quickly focus on the values near the target bounding box location by expanding the probability values of the values near the target bounding box location. bounding box location, allowing the network to quickly focus on values near the target bounding box. By modeling the bounding box locations as general distributions, DFL can provide more informative and accurate estimates of the bounding box, and the idea is to use a cross-entropy function to optimize the probabilities of the left and right locations near the label y, and to focus the network distribution near the labeled values. The global minimum solutions for DFL are $S_i = \frac{y_i - y}{y_{i+1} - y_i}$ and $S_{i+1} = \frac{y - y_i}{y_{i+1} - y_i}$, which ensures that the estimated regression objective $\bar{y}$ is infinitely close to the corresponding label y, thus ensuring its correctness as a loss function.

Category imbalance is a prevalent challenge in target detection tasks, where some categories may possess an abundance of samples, while others may be underrepresented. The DFL function is a more effective solution in addressing category imbalance by incorporating category distribution information, thus enhancing detection performance for categories with fewer samples. Additionally, the design of DFL endows it with resilience against misclassification, as it imposes a higher loss on samples that are challenging to categorize, thereby directing the model's attention to these samples and reducing the likelihood of misclassifications.

In this article, we have implemented the DFL loss function within the Head structure to compute regression values, rather than directly obtaining the regression values. We employ two independent convolutions for each feature layer to adjust the number of channels, enabling us to derive the regression values for the prediction frame corresponding to each feature point target individually.

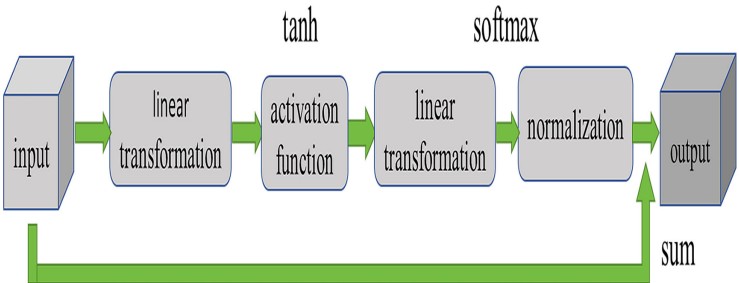

**Figure 3 Structure of attention mechanism.** 

## Attention mechanism

Inspired by the way humans process visual information, attention mechanisms have been developed to enable computers to concentrate their attention on the most informative aspects of the input signal. Attention mechanisms originally employed in machine translation (*Lin et al., 2017b*), which have been proven to enhance the understanding of contextual relationships. Subsequently, this technique has found widespread application in computer vision, yielding remarkable results. Within neural networks, attention mechanisms can be integrated into the spatial dimension, the channel dimension (*Bahdanau, Cho & Bengio, 2014*), or a combination of both, thereby providing models with greater flexibility in handling complex input data and enhancing their performance in tasks like image classification, target detection, and image segmentation.

Spatial attention (*Hu, Shen & Sun, 2018*) is a commonly used attention mechanism. When introduced in the spatial dimension, it allows the model to learn and assign different attention weights to various regions of an image. Consequently, the model can prioritize important regions of the image, leading to improved target detection and recognition. Conversely, channel attention (*Wang et al., 2020*) introduces the attention mechanism in the channel dimension, enabling the model to allocate varying attention weights to feature channels. This approach allows the model to focus more on feature channels that are more relevant to the task, reducing the response to irrelevant information and, in turn, enhancing the model's overall performance. Hybrid attention mechanisms that combine spatial and channel dimensions also exist (*Zhang & Sabuncu, 2018*). By integrating attention mechanisms into both spatial and channel dimensions, these models can simultaneously focus on different regions and feature channels within an image (*Guan et al., 2023*). This comprehensive approach enables the model to capture important information in the image more effectively. Adhering to the standards set by scientific publications, the integration of attention mechanisms has revolutionized the field of computer vision and substantially improved its capabilities in various visual recognition tasks.

Currently, widely used attention mechanisms include SE, CA (*Hou, Zhou & Feng, 2021*), CBAM (*Woo et al., 2018*), *etc.*, all of which acquire critical information about the target on a global scale. However, these methods often come with high computational complexity and significant memory consumption. To enhance the model's feature extraction capabilities while avoiding excessive computational burden, this article

introduces a new attention mechanism by modifying SE. As depicted in Fig. 3, this attention structure takes an input tensor, initially mapping the input data through a linear layer to a hidden layer, followed by a non-linear transformation using the tanh activation function. Subsequently, the output of the hidden layer is mapped through another linear layer to obtain attention weights, which are then normalized using the softmax function to derive the final attention weights. Finally, the input features are multiplied by the attention weights and summed to obtain the ultimate output.

As illustrated in Fig. 3, this attention structure takes an input tensor and produces a weighted sum along with the corresponding attention weights. In the forward propagation process, we initiate by calculating the hidden representation through a linear layer followed by a nonlinear activation function. Subsequently, attention weights are computed through another linear layer, combined with a softmax activation function. Finally, these attention weights are multiplied with the input tensor to yield the weighted sum.

Under the standards set by scientific publications, these enhancements contribute to a more efficient and powerful feature extraction mechanism, ensuring the model's improved performance without overburdening the network with excessive computational demands.

Firstly, our attention mechanism can more accurately capture key information and utilize it more effectively. Compared to traditional SE methods, our design allows for finer adjustments of attention weights, enabling the model to focus more intensely on features that have a greater impact on the task.

Secondly, our attention mechanism performs better when handling information of different scales. Through clever design, we enable the model to comprehensively understand input data at various levels, thereby enhancing the model's capability to process complex data.

Finally, our attention mechanism demonstrates better convergence and stability during training. Through thoughtful design, we successfully mitigate the risk of overfitting or underfitting in the model during training, thereby improving the model's generalization ability.

# EXPERIMENT

The hardware environment used in this article is 64-bit Windows 10, the GPU used is NVIDIA Tesla V100-SXM2 32GB, CUDN is version 10.0, CUDNN is version 10.1, and the learning framework is Pytorch 1.2.0. The model is set to an initial learning rate of 1e−2, trained by default for 300 epochs, with the trunk frozen for the first 50 epochs, using stochastic gradient descent (SGD) optimizer.

There are 936 remote sensing images in the RSOD (*Long et al., 2017*) dataset, the RSOD dataset includes four categories: aircraft, oil tank, overpass, and playground, the RSOD dataset totals 446 aircraft images, 189 playground images, 4,993 aircraft samples, 191 playground samples, 176 overpass images, 180 overpass samples, 165 oil drum images, and 1,586 oil drum samples.

## Evaluation index

The experiment uses precision rate and average precision mean (mAP@0.5) as the performance evaluation criteria for evaluating the target detection methods. The precision rate and recall are calculated as shown below:

$$P = \frac{TP}{TP + FP} \tag{6}$$

$$R = \frac{TP}{TP + FN} \tag{7}$$

$$AP = \sum_{i=1}^{n-1} (r_{i+1} - r_i)\, P_{inter}(r_i + 1) \tag{8}$$

$$mAP = \frac{\sum_{i=0}^{n} AP_i}{n} \tag{9}$$

$P$ is precision, TP represents the number of positive samples correctly identified as positive, FP represents the number of negative samples incorrectly identified as positive. Precision $P$ is the ratio of true positives to the total number of samples identified as positive by the model, including both true positives and false positives. It provides the accuracy of the model in identifying positives. A higher precision value indicates that the model more accurately identifies positives, while a lower precision value suggests that the model may incorrectly classify negatives as positives. The detailed formula for precision $P$ is shown in Eq. (6).

R is recall, which measures the proportion of positive instances correctly identified by the model among all actual positive instances. FN represents the number of positive samples incorrectly identified as negative. Recall is the ratio of true positives to the total number of actual positives, including both true positives and false negatives. It evaluates the model's ability to capture all positives in the dataset. A higher recall value indicates that the model is better at identifying positives, while a lower recall value suggests that the model may miss some positives. The calculation formula for recall R is shown in Eq. (7).

AP is average precision, where n is the total number of true positives, $r_i$ is the number of true positives in the top i predicted results returned by the model, ($P_{inter}(r_{i+1})$) is the interpolated precision when the recall is $r_{i+1}$. Average precision evaluates the model's performance based on the relationship between the predicted results returned by the model and the ground truth labels, as well as the precision values within each recall interval. The calculation formula for average precision AP is shown in Eq. (8).

$AP_i$ refers to the average precision of the ith class detection target, which is the area under the PR curve. n represents the number of categories. mAP, the mean average precision, is the average of the average precision of all categories, providing a comprehensive evaluation of the model's performance across multiple categories. The calculation formula for mAP is shown in Eq. (9).

## Ablation experiment

Figure 4 demonstrates the F1 score status of this experiment. To verify the effectiveness of adding the DFL loss function, the improved multi-scale feature extraction module SSN,

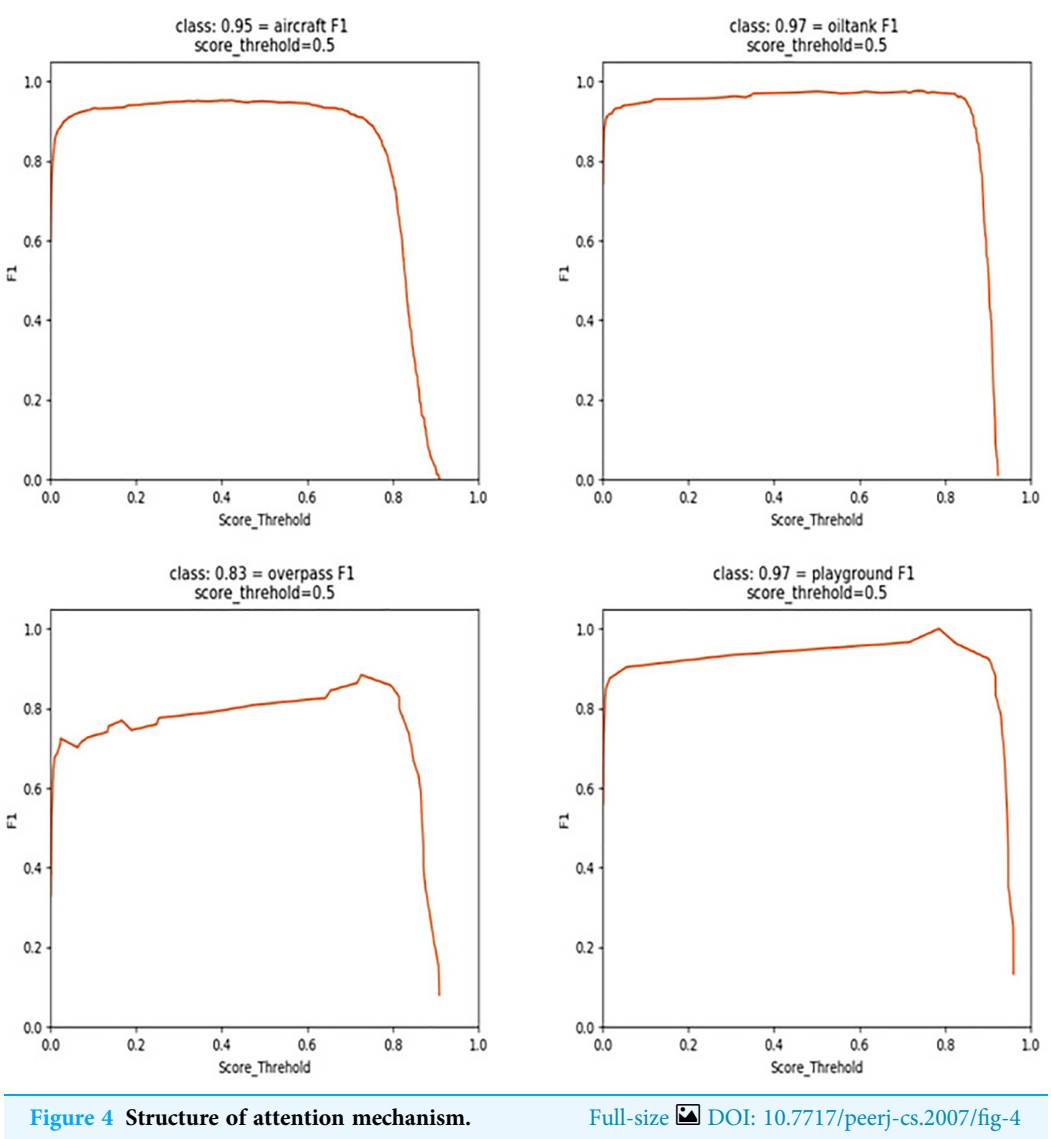

**Figure 4 Structure of attention mechanism.**

and the new attention mechanism, this article chooses YOLOv5s as the baseline model and evaluates the impact of different modules and methods on the target detection performance when they are combined through the ablation experiments under the same experimental conditions.

In Table 1, The baseline model is YOLOv5s, achieves precision, recall, and mAP scores of 90.61%, 97.92%, and 91.93%, respectively, with a parameter count of 7.46 MB 1. Upon integrating the DFL loss function into the baseline model, the mAP increases from 91.93% to 93.52%, accompanied by a reduction in parameters, indicating a significant enhancement in boundary box detection accuracy. Further improvement is observed when incorporating the SSN structure, resulting in a heightened detection accuracy of 93.96%, with a reduced parameter count of 7.15 M. Subsequent integration of the new VAM attention mechanism leads to an mAP increase to 92.02%. In subsequent grouped comparative experiments, the performance is significantly enhanced when both DFL and

**Table 1 Ablation experiments on the RSOD dataset.**

| Algorithm | P (%) | R (%) | mAP (%) | Parameters (MB) |
|---|---|---|---|---|
| Baseline (YOLOv5s) | 90.61 | 97.92 | 91.93 | 7.46 |
| Baseline+DFL | 91.34 | 96.81 | 93.52 | 7.38 |
| Baseline+NCSP | 89.92 | 98.21 | 93.96 | 7.15 |
| Baseline+VAM | 92.32 | 96.14 | 92.02 | 7.52 |
| Baseline+DFL+SSN | 92.55 | 96.52 | 94.23 | 7.41 |
| Baseline+DFL+VAM | 92.83 | 95.82 | 93.75 | 7.39 |
| Baseline+SSN+VAM | 91.53 | 96.81 | 93.56 | 7.24 |
| Baseline+DFL+NCSP+VAM | 93.15 | 92.76 | 95.44 | 7.26 |

**Table 2 Effectiveness of different algorithms on RSOD dataset.**

| Model | Airplane | Oil tank | Playground | Overpass | mAP (%) |
|---|---|---|---|---|---|
| SSD | 87.3 | 92.5 | 85.3 | 80.4 | 86.4 |
| YOLOV4 | 90.1 | 93.6 | 88.4 | 84.3 | 89.1 |
| FasterR-CNN | 92.3 | 95.7 | 89 | 83.6 | 90.2 |
| YOLOv5s | 94.2 | 96.4 | 93.7 | 83.3 | 91.9 |
| YOLOV7 | 96.9 | 99.6 | 99.4 | 89.2 | 96.1 |
| Ours | 96.2 | 99.7 | 99.1 | 85.7 | 95.4 |

SSN are applied, achieving an mAP of 94.23%. Similarly, when both DFL and VAM are applied, there is a slight performance improvement, reaching an mAP of 94.23%, with a reduced parameter count of 7.39 MB. The best performance is achieved when DFL, SSN, and VAM are all applied simultaneously, yielding a precision of 93.15% and an mAP of 95.44%, with a parameter count of 7.26 MB. Compared to the baseline, there is a 3.51% improvement in mAP, along with a 0.2 reduction in parameter count, demonstrating optimization in both accuracy and parameter usage. These results affirm the effectiveness of the algorithmic improvements made in this study.

Table 2 provides a comparison of the detection performance between our proposed method and other classical algorithms on the RSDO dataset to further validate the effectiveness of our approach. The table presents the detection accuracy of various algorithms across different categories in the RSOD dataset. Noticeable performance differences among different models are observed across different categories. For instance, in the "Oil tank" category, both YOLOv7 and our model exhibit notably higher performance compared to other models, achieving 99.6% and 99.7%, respectively, while in the "Overpass" category, YOLOv7 and our model demonstrate relatively lower performance, with accuracies of 89.2% and 85.7%, respectively. Upon observing the overall average precision (AP) results, it is noted that YOLOv7 and our model slightly outperform other models, achieving 96.1% and 95.4% AP, respectively, indicating their competitiveness in overall performance. Conversely, SSD and Faster R-CNN exhibit lower performance, with accuracies of 86.4% and 90.2%, respectively. YOLOv5s performance

| Table 3 Comparative experiments of different attention mechanisms. | | |
|---|---|---|
| Model | Parameters/MB | mAP/% |
| SE | 7.55 | 91.6 |
| CBAM | 7.48 | 91.1 |
| ECA | 7.55 | 90.5 |
| CA | 7.44 | 92.1 |
| VAM | 7.52 | 92.2 |

falls between these two extremes, with an accuracy of 91.9%. These results reflect performance discrepancies among different models across various categories. From Table 2, it is evident that YOLOv7 achieves the highest detection accuracy; however, YOLOv7 also possesses a deeper network, resulting in a higher parameter count compared to YOLOv5. Considering the application scenarios' requirements, we ultimately select YOLOv5 as the baseline for algorithmic improvements.

To assess the effectiveness of the newly designed attention mechanism in this article and analyze the performance of various attention mechanisms within the algorithm, SE, CBAM, ECA, CA, and the VAM introduced in this article are chosen for comparative experiments. In each experiment, a specific attention mechanism module is employed in the network's feature extraction section of the YOLOv5s architecture. To ensure the validity of the comparison experiments, the different attention mechanism modules are placed at the same positions in the network structure while maintaining the network's remaining configurations. Additionally, the same loss function is employed for model training and testing. The results of the comparison experiments for attention mechanisms are detailed in Table 3.

Table 3 provides data comparing the performance of different attention mechanisms (SE, CBAM, ECA, CA, VAM) applied to object detection models. By observing the relationship between model parameters and average precision (mAP), several important conclusions can be drawn.

Firstly, from the perspective of model parameters, it can be observed that different attention mechanisms have varying degrees of impact on the model's parameter count. Specifically, the Channel Attention (CA) model has the lowest parameter count (7.44 MB), while the Efficient Channel Attention (ECA) model has the highest parameter count (7.55 MB). This indicates that different attention mechanisms have different effects on model complexity, and selecting the appropriate attention mechanism can minimize the model's parameter count while maintaining performance.

Secondly, from the perspective of average precision (mAP), it can be observed that different attention mechanisms have different effects on model performance. The VAM model exhibits the highest average precision (92.2%), while the ECA model shows the lowest average precision (90.5%). This suggests that selecting the appropriate attention mechanism can significantly impact the model's performance, thereby affecting the accuracy and stability of object detection.

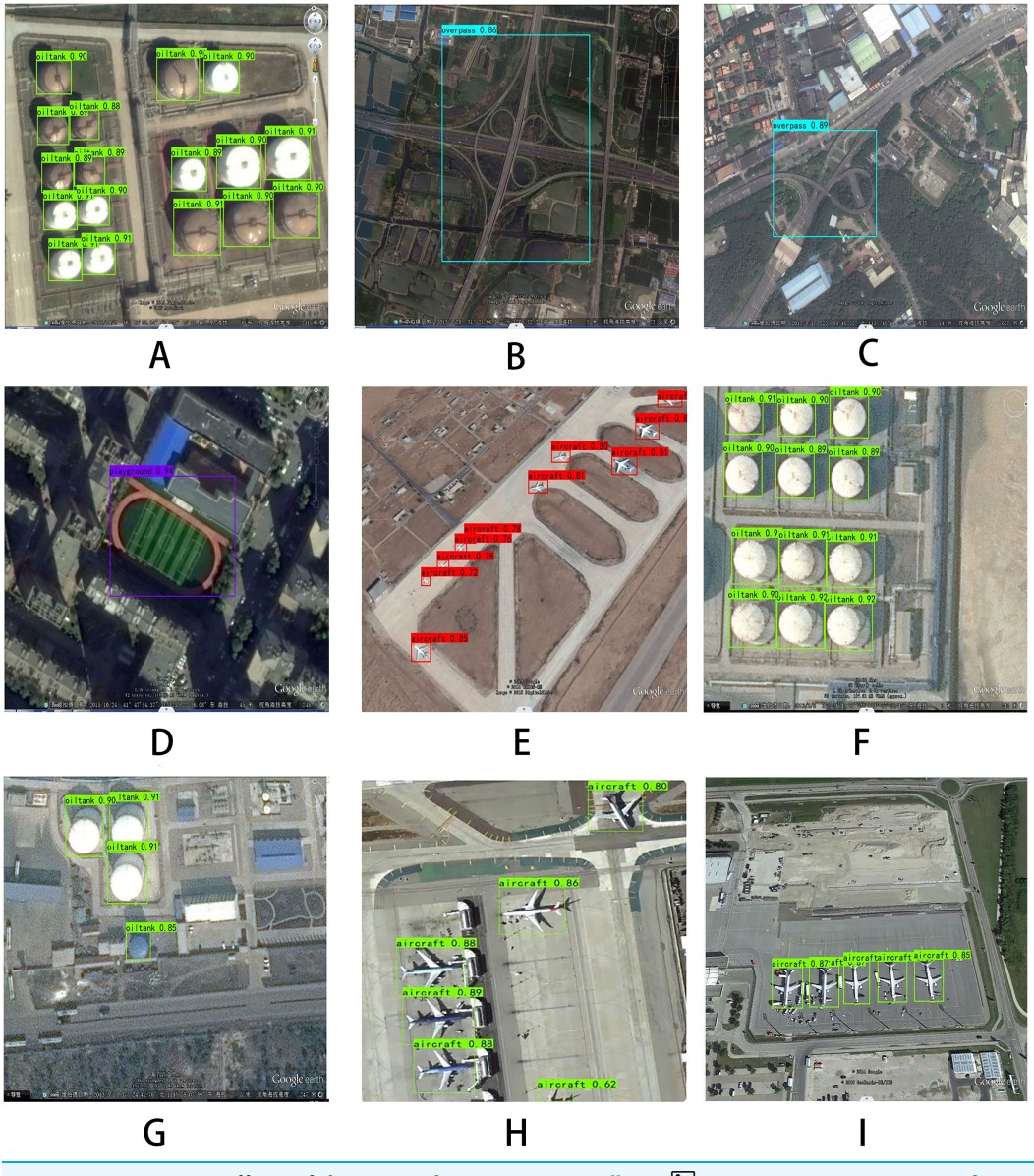

**Figure 5 Detection effects of the RSOD dataset.**

From Table 3, it can be seen that VAM has better detection performance in this model, with a 0.6 improvement compared to SE. This implies that VAM helps the convolutional network better capture image features while also performing well in model parameters. Compared to SE, it reduces computational complexity, making it more lightweight and suitable for deployment on resource-constrained devices. Therefore, it can be inferred from Table 3 that the attention mechanism VAM designed in this article has a significant impact.

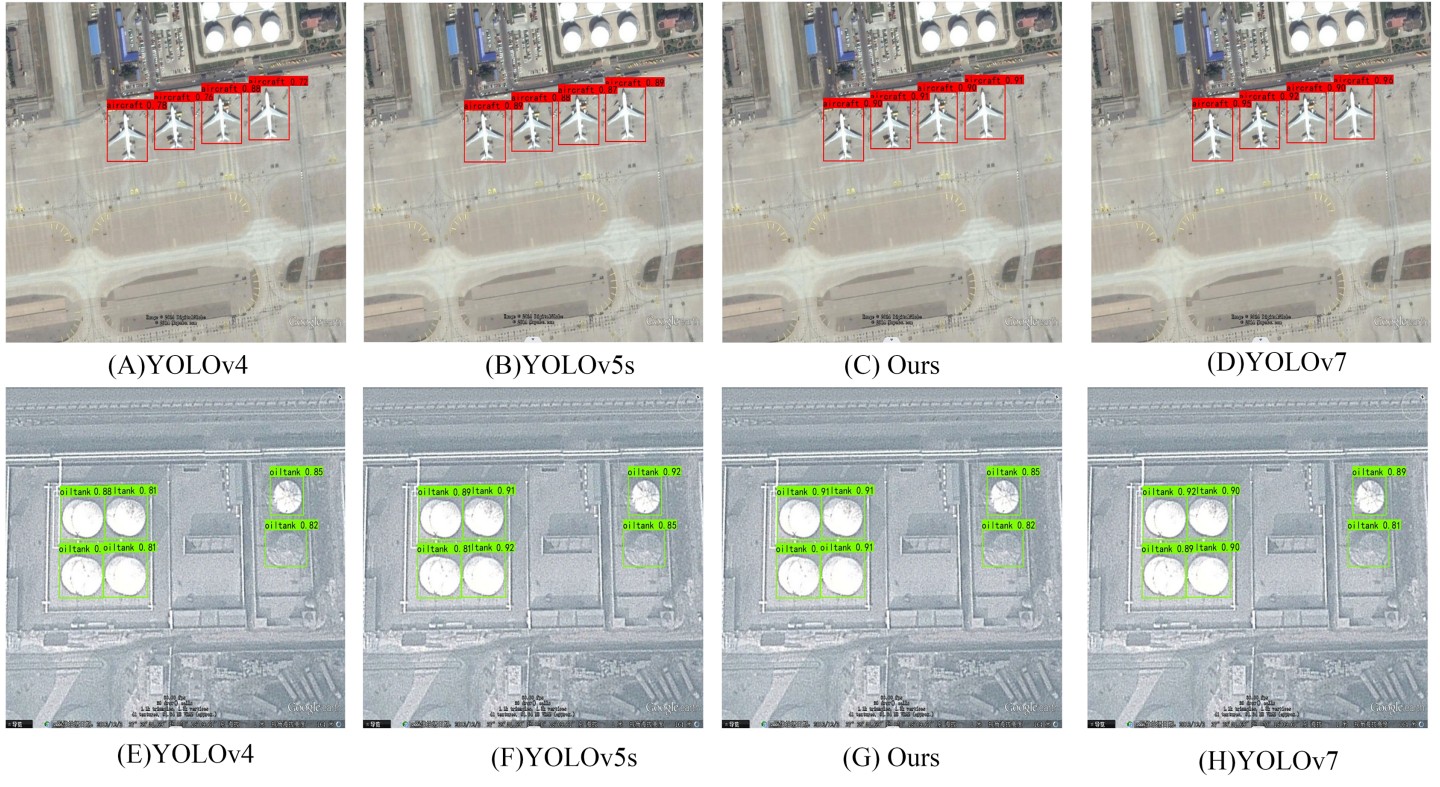

**Figure 6  Comparative status of RSOD dataset on different algorithms.**                

## Result visualization

In this article, multiple images are taken from the RSOD dataset, and the values in the figure represent the confidence level of this detection effect, which refers to the size of the probability of detecting the authenticity of the target. The detection effect of the model is shown in Fig. 5.

To thoroughly investigate the comparative detection performance before and after the implementation of our improved algorithm, we selected an image from the RSOD dataset containing small targets such as aircraft and oil tanks. We conducted detection using classic algorithms YOLOv4, YOLOv5s, our improved algorithm, and YOLOv7. The detection results are depicted in Fig. 6.

Primarily, it is evident from the images that the detection accuracy of YOLOv4 is comparatively low, performing the worst among the models considered. This might be attributed to factors such as the architectural design or optimization techniques of YOLOv4. A clear observation from the image compositions in Fig. 6 indicates that the original algorithm exhibits poor detection of small targets such as oil tanks and aircraft.

Furthermore, our modified models, through improvements in loss functions, network architecture, and the addition of attention mechanisms, show a certain degree of improvement in detection accuracy compared to YOLOv5s. This suggests that modifications made to the model architecture and training process have positively influenced its performance. Comparison with YOLOv7 demonstrates that our modified

model achieves detection results comparable to YOLOv7. This indicates that while YOLOv7 may contain more advanced features or optimizations, our model can achieve similar performance levels through customized modifications.

Through modifications such as altering the feature extraction network structure, introducing new DFL loss functions, and integrating better-performing attention mechanisms, significant enhancements have been achieved. As depicted in the image compositions in Fig. 6, our algorithm can comprehensively detect objects, with a notable improvement in the accuracy of detection boxes. These improvement strategies make the algorithm more suitable for feature extraction in small targets, effectively addressing the issue of target scale differences, particularly enhancing performance in small target detection. Additionally, our algorithm pays more attention to difficult-to-classify targets, further enhancing detection accuracy. These improvements make remote sensing image object detection more reliable and precise. The refined algorithm should more accurately identify and locate these targets, reducing instances of missed detections, thereby enhancing the reliability and accuracy of remote sensing object detection.

## CONCLUSION

This article addresses the challenges of small object detection and inadequate recognition accuracy in aerial images and proposes an improved algorithm based on YOLOv5s. Specifically, our contributions can be summarized in three aspects:

Firstly, we introduce the DFL loss function. This loss function, by specially processing pixels at boundaries during neural network training, enhances the model's accuracy and robustness in recognizing objects at boundaries. By designing the DFL loss function, we enhance the robustness and accuracy of the model in small object detection.

Secondly, we introduce a novel Small-Scale Network (SNN) module to improve the neural network. This module enhances the model's representational capacity by adding convolutional layers, using smaller convolutional kernels, and introducing residual connections. It better captures rich, abstract features of the data, thereby improving the model's performance, particularly in small object detection.

Lastly, we introduce an improved attention mechanism structure. This structure, by appropriately allocating weights, effectively promotes the interaction and fusion of information between different contexts, optimizing the intrinsic structure of information at different levels. This innovation aids in enhancing the model's perception and inference capabilities.

We comprehensively evaluate and analyze the improved model. Through comparative experiments and ablation studies, we found that our proposed improved algorithm achieved a significant improvement in mAP (mean Average Precision) by 3.51 compared to the baseline, while also reducing the model parameter count by 0.2. Compared to other common network models, our model improvement also demonstrates certain advantages.

In summary, the proposed improved algorithm makes significant advancements in the field of small object detection, enhancing detection accuracy and model lightweightness. It is poised to provide more powerful tools and performance for aerial image processing in practical applications.

### Funding

This work was funded by the National Natural Science Foundation of China (12201185). The funders had no role in study design, data collection and analysis, decision to publish, or preparation of the manuscript.

### Grant Disclosures

The following grant information was disclosed by the authors:
National Natural Science Foundation of China: 12201185.

### Competing Interests

The authors declare that they have no competing interests.

### Author Contributions

- PengLei Zhang conceived and designed the experiments, analyzed the data, performed the computation work, prepared figures and/or tables, and approved the final draft.
- Yanhong Liu performed the experiments, authored or reviewed drafts of the article, and approved the final draft.

### Data Availability

The code is available at Zenodo: zhangpenglei. (2024). zhangpenglei/Target-Detection: Target-DetectionRelease (Version v1). Zenodo. https://doi.org/10.5281/zenodo.10722012.

The RSOD dataset is available at GitHub: https://github.com/RSIA-LIESMARS-WHU/RSOD-Dataset-.

The dataset needs to be converted to match the appearance of this network, and the conversion file VOC_ Annotation.py.

### Supplemental Information

Supplemental information for this article can be found online at http://dx.doi.org/10.7717/peerj-cs.2007#supplemental-information.

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
