# Peer review of "A small target detection algorithm based on improved YOLOv5 in aerial image"

_PeerJ Computer Science, doi:10.7717/peerj-cs.2007_

## Round 0.1 · original submission · Major Revisions

Please carefully go through the reviews and submit a new draft.

Apart from the English grammar, please address the reviewer-2 comments. Feel free to justify if a reviewer's comment doesn't apply to your paper.

Looking forward to the new version of the paper.

**Language Note:** The review process has identified that the English language must be improved. PeerJ can provide language editing services - please contact us at copyediting@peerj.com for pricing (be sure to provide your manuscript number and title). Alternatively, you should make your own arrangements to improve the language quality and provide details in your response letter. – PeerJ Staff

·

Basic reporting

1. there are so many mistakes in paper. for example,"Zhu et al. Zhu et al. (2021)","Wang Wang et al. (2023)and Zhu Zhu et al. (2021)" in page 2. "he YOLOv5"in page3.

2.The article must be written in English and must use clear, unambiguous, technically correct text. The article must conform to professional standards of courtesy and expression.

3. Figure 3 is incomprehensible

Experimental design

1.The content of the chart does not match the title, for example table2 and table 3

2. the mAP is mAP0.5 or mAP0.95?

3. The innovation of this paper is pool

Validity of the findings

1.there are not add enough latest methods for comparison, such as yolov7,yolov8, and the other state of the art methods.

Additional comments

The detection method is presented based on yolov5, why not yolov7 or yolov8? The improvement based on yolov7 or yolov8 may perform better.

·

Basic reporting

1.English language skills as well as expression should be improved to ensure that all readers can clearly understand your essay. There is a typo in line 99 and it is recommended to double check the whole text.
2.The descriptions provided for the figures in the manuscript, such as Figure 1 and Figure 2, are rather succinct, potentially presenting difficulties for novice readers in grasping the individual components' functions and underlying principles comprehensively.
3.The article lacks a more detailed description of the principles behind the proposed modules, such as the C2F module.
4.It is recommended to optimize the picture. For example, the lines in Figure 1 are a little messy.
5.Try to explain what the variables in formula (2) represent and what they mean.
6.In the "Evaluation Index" section, it is proposed to add the formula for the calculation of AP.

Experimental design

7.The official YOLOv5 parameter count of YOLOv5s is only 7.2M, why is it 48.06M in the experimental part of the paper?Why is one of the baseline parameter counts of Table 1 and Table 2 48.06M and the other 62.41M?Does the parameter count of the model transform with the change of the dataset? Please give a reasonable explanation.
8.In Table 1 of the "Ablation Experiment" section, are each module (DFL, NCSP, VAM) added based on the Baseline, or based on the previous module? Why are there no test results for pairwise module combinations?
9.Table 2 is the result on which data set? Is the header of Table 2 incorrectly described? The header of Table 2 does not seem to match the description in the text.
10.There seems to be no comparative experiment between DFL and other loss functions in the article. Is there a more effective loss function? It is recommended to add a comparative experiment with the loss function.
11.Table 3 only compares SSD, YOLOV4, and Faster R-CNN, which are classic algorithms from a few years ago, but does not compare some newer algorithms in the past two years (such as YOLOV7, YOLOx, etc.), and it does not even compare with the original algorithm YOLOv5. It is recommended to add comparative experiments.
12.Please confirm whether the expressions in lines 275-279 are correct. The commonly referred mAP is used to measure the detection results of multiple categories, rather than a single category as mentioned in the article.
13.In the "Result Visualization" section, there are only comparison results with the original algorithm YOLOv5s, but no comparison results with other algorithms. It is recommended to add comparative experiments and analysis of some classic algorithms and some newer algorithms.

Validity of the findings

no comment

---

## Round 0.2 · accepted · Accept

Thanks for addressing all the comments of the reviewers. I think your paper is in very good shape now. I have reviewed the revised version and the overall quality is much improved. The manuscript is now ready for publication. PeerJ staff may reach out to you for any other remaining publication tasks.